# SPREAD DIVERGENCES

## ABSTRACT

For distributions $p$ and $q$ with different support, the divergence $D(p||q)$ generally will not exist. We define a spread divergence $\tilde{D}(p||q)$ on modified $p$ and $q$ and describe sufficient conditions for the existence of such a divergence. We give examples of using a spread divergence to train implicit generative models, including linear models (Principal Components Analysis and Independent Components Analysis) and non-linear models (Deep Generative Networks).

## 1 INTRODUCTION

A divergence $D(p||q)$ (see, for example Dragomir (2005)) is a measure of the difference between two distributions $p$ and $q$ with the property

$$D(p||q) \geq 0, \quad \text{and} \quad D(p||q) = 0 \quad \Leftrightarrow \quad p = q \qquad (1)$$

Some of our results are specific to the $f$-divergence, defined as

$$D_f(p||q) = \mathbb{E}_{q(x)}\left[ f\left( \frac{p(x)}{q(x)} \right) \right] \qquad (2)$$

where $f(x)$ is a convex function with $f(1) = 0$. An important special case of an $f$-divergence is the well-known Kullback-Leibler divergence $KL(p||q) = \mathbb{E}_{p(x)}\left[ \log \frac{p(x)}{q(x)} \right]$ which is widely used to train models using maximum likelihood. We are interested in situations in which the supports of the two distributions are different, $\text{supp}(p) \neq \text{supp}(q)$. In this case the divergence may not be defined. For example, for $p(x)$ being an empirical data distribution on continuous dataset $x_1, \ldots, x_N$, $p(x) = \frac{1}{N}\sum_{n=1}^{N} \delta(x - x_n)$ where $\delta(\cdot)$ is the Dirac Delta function. For a model $q(x)$ with support $\mathbb{R}$, then $KL(q||p)$ is not formally defined. This is a challenge since implicit generative models of the form $q(x) = \int \delta(x - g_\theta(z)) p(z)dz$ only have limited support; in this case maximum likelihood to learn the model parameter $\theta$ is not available and alternative approaches are required – see Mohamed & Lakshminarayanan (2016) for a recent survey.

## 2 SPREAD DIVERGENCES

The aim is, from $q(x)$ and $p(x)$ to define new distributions $\tilde{q}(y)$ and $\tilde{p}(y)$ that have the same support[1]. Using the notation $\int_x$ to denote integration $\int (\cdot) dx$ for continuous $x$, and $\sum_{x \in \mathcal{X}}$ for discrete $x$ with domain $\mathcal{X}$, we define a random variable $y$ with the same domain as $x$ and distributions

$$\tilde{p}(y) = \int_x p(y|x)p(x), \qquad \tilde{q}(y) = \int_x p(y|x)q(x) \qquad (3)$$

where $p(y|x)$ is a 'noise' process designed to 'spread' the mass of $p$ and $q$ such that $\tilde{p}(y)$ and $\tilde{q}(y)$ have the same support. For example, if we use a Gaussian $p(y|x) = \mathcal{N}\left(y|x, \sigma^2\right)$, then $\tilde{p}$ and $\tilde{q}$ both have support $\mathbb{R}$. We therefore use noise with the property that, despite $D(p||q)$ not existing, $D(\tilde{p}||\tilde{q})$ does exist and we define the Spread Divergence

$$\tilde{D}(p||q) \equiv D(\tilde{p}||\tilde{q}) \qquad (4)$$

Note that this satisfies the divergence requirement $\tilde{D}(p||q) \geq 0$. The second requirement, $\tilde{D}(p||q) = 0 \Leftrightarrow p = q$, is guaranteed for certain 'noise' processes, as described in section(2.1).

---

[1]For simplicity, we use univariate $x$, with the extension to the multivariate setting being straightforward.

Spread divergences have many potential applications. For example, for a model $q(x|\theta)$ with parameter $\theta$ and empirical data distribution $p(x)$, maximum likelihood training corresponds to minimising $\text{KL}(p||q)$ with respect to $\theta$. However, for implicit models, the divergence $\text{KL}(p||q)$ does not exist. However, if a spread divergence exists, provided that the data is distributed according the model $p(x) = q(x|\theta_0)$ for some unknown parameter $\theta_0$, the spread divergence $\text{D}(p(x)||q(x|\theta))$ has a minimum at $\theta = \theta_0$. That is (for identifiable models) we can correctly learn the underlying data generating process, even when the original divergence is not defined.

### 2.1 Noise Requirements for a Spread Divergence

Our main interest is in using noise to define a new divergence in situations in which the original divergence $\text{D}(p||q)$ is itself not defined. For discrete variables $x \in \{1, \ldots, n\}$, $y \in \{1, \ldots, n\}$, the noise $P_{ij} = p(y = i|x = j)$ must be a distribution $\sum_i P_{ij} = 1$, $P_{ij} \geq 0$ and

$$\sum_j P_{ij} p_j = \sum_j P_{ij} q_j \quad \forall i \quad \Rightarrow \quad p_j = q_j \quad \forall j \tag{5}$$

which is equivalent to the requirement that the matrix $P$ is invertible, see appendix(B). There is an additional requirement that the spread divergence exists. In the case of $f$-divergences, the spread divergence exists provided that $\tilde{p}$ and $\tilde{q}$ have the same support. This is guaranteed if

$$\sum_j P_{ij} p_j > 0, \quad \sum_j P_{ij} q_j > 0 \quad \forall i \tag{6}$$

which is satisfied if $P_{ij} > 0$. In general, therefore, there is a space of noise distributions $p(y|x)$ that define a valid spread divergence. The 'antifreeze' method of Furmston & Barber (2009) is a special form of spread noise to define a valid Kullback-Leibler divergence (see also Barber (2012)).

For continuous variables, in order that $\tilde{\text{D}}(p||q) = 0 \Rightarrow p = q$, the noise $p(y|x)$, with $\dim(Y) = \dim(X)$ must be a probability density and satisfy

$$\int p(y|x)p(x)dx = \int p(y|x)q(x)dx \quad \forall y \in Y \quad \Rightarrow \quad p(x) = q(x) \quad \forall x \in X \tag{7}$$

This is satisfied if there exists a transform $p^{-1}$ such that

$$\int p^{-1}(x'|y)p(y|x)dy = \delta(x' - x) \tag{8}$$

where $\delta(\cdot)$ is the Dirac delta function. As for the discrete case, the spread divergence exists provided that $\tilde{p}$ and $\tilde{q}$ have the same support, which is guaranteed if $p(y|x) > 0$. A well known example of such an invertible integral transform is the Weierstrass Transform $p(y|x) = \mathcal{N}(y|x, \sigma^2)$, which has an explicit representation for $p^{-1}$. In general, however, we can demonstrate the existence of a spread divergence without the need for an explicit representation of $p^{-1}$. As we will see below, the noise requirements for defining a valid spread divergence such that $\text{D}(\tilde{p}||\tilde{q}) = 0 \Leftrightarrow p = q$ are analogous to the requirements on kernels such that the Maximum Mean Discrepancy $\text{MMD}(p, q) = 0 \Leftrightarrow p = q$, see Sriperumbudur et al. (2011) and Sriperumbudur et al. (2012).

## 3 Stationary Spread Divergences

Consider stationary noise $p(y|x) = K(y - x)$ where $K(x)$ is a probability density function with $K(x) > 0$, $x \in \mathbb{R}$. In this case $\tilde{p}$ and $\tilde{q}$ are defined as a convolution

$$\tilde{p}(y) = \int K(y - x)p(x)dx = (K * p)(y), \quad \tilde{q}(y) = \int K(y - x)q(x)dx = (K * q)(y) \tag{9}$$

Since $K > 0$, $\tilde{p}$ and $\tilde{q}$ are guaranteed to have the same support $\mathbb{R}$. A sufficient condition for the existence of the Fourier Transform $\mathcal{F}\{f\}$ of a function $f(x)$ for real $x$ is that $f$ is absolutely integrable. All distributions $p(x)$ are absolutely integrable, so that both $\mathcal{F}\{p\}$ and $\mathcal{F}\{q\}$ are guaranteed to exist. Assuming $\mathcal{F}\{K\}$ exists, we can use the convolution theorem to write

$$\mathcal{F}\{\tilde{p}\} = \mathcal{F}\{K\}\mathcal{F}\{p\}, \quad \mathcal{F}\{\tilde{q}\} = \mathcal{F}\{K\}\mathcal{F}\{q\} \tag{10}$$

Hence, we can write

$$\mathrm{D}(\tilde{p}||\tilde{q}) = 0 \Leftrightarrow \tilde{p} = \tilde{q} \Leftrightarrow \mathcal{F}\{K\}\mathcal{F}\{p\} = \mathcal{F}\{K\}\mathcal{F}\{q\} \Leftrightarrow \mathcal{F}\{p\} = \mathcal{F}\{q\} \Leftrightarrow p = q \quad (11)$$

where we used the invertibility of the Fourier transform and assumed that $\mathcal{F}\{K\} \neq 0$, or equivalently[2], $\mathcal{F}\{K\} > 0$. Hence, provided that $K(x) > 0$ and $\mathcal{F}\{K\} > 0$ then $K(x)$ defines a valid spread divergence. Note that other transforms have a corresponding convolution theorem[3] and the above derivation holds, with the requirement that the corresponding transform of $K(x)$ is non-zero. As an example of such a noise process, consider Gaussian noise,

$$K(y - x) = \frac{1}{\sqrt{2\pi\sigma^2}} e^{-\frac{1}{2\sigma^2}(y-x)^2} \quad (12)$$

leading to a positive Fourier Transform:

$$\mathcal{F}\{K\}(\omega) = \frac{1}{\sqrt{2\pi\sigma^2}} \int_{-\infty}^{\infty} e^{i\omega x} e^{-\frac{1}{2\sigma^2}x^2} dx = e^{-\frac{\sigma^2\omega^2}{2}} > 0 \quad (13)$$

Similarly, for Laplace noise $K(x) = \frac{1}{2b} e^{-\frac{1}{b}|x|}$

$$p(y|x) = K(y - x) = \frac{1}{2b} e^{-\frac{1}{b}|y-x|}, \qquad \mathcal{F}\{K\}(\omega) = \sqrt{\frac{2}{\pi}} \frac{b^{-1}}{b^{-2} + \omega^2} > 0 \quad (14)$$

Since $K > 0$ and $\mathcal{F}\{K\} > 0$, this also defines a valid spread divergence over $\mathbb{R}$.

## 3.1 INVERTIBLE MAPPINGS

Consider $p(y|x) = K(y - f(x))$ for strictly monotonic $f$. Then, using the change of variables

$$\tilde{p}(y) = \int K(y - z)p_z(z)dz, \qquad p_z(z) = p_x(f^{-1}(z))\left(|J\left(x = f^{-1}(z)\right)|\right)^{-1} \quad (15)$$

where $J$ is the Jacobian of $f$. For distributions $p(x)$ with bounded domain, for example $x \in [0,1]$, we can use a logit function, $f(x) = -\log\left(x^{-1} - 1\right)$, which maps the interval $[0,1]$ to $\mathbb{R}$. Using then, for example, Gaussian spread noise $p(y|x) = \mathcal{N}\left(y|f(x), \sigma^2\right)$, both $\tilde{p}(y)$ and $\tilde{q}(y)$ have support $\mathbb{R}$. If $\mathrm{D}(\tilde{p}||\tilde{q})$ is zero then $p = q$ on the domain $[0,1]$.

## 3.2 MAXIMISING THE SPREAD

From the data processing inequality (see appendix(A)), spread noise will always decrease the $f$-divergence $\mathrm{D}_f(\tilde{p}(y)||\tilde{q}(y)) \leq \mathrm{D}_f(p(x)||q(x))$. If we are to use a spread divergence to train a model, there is the danger that adding too much noise may make the spreaded empirical distribution and spreaded model distribution so similar that it becomes difficult to numerically distinguish them, impeding training. In general, therefore, it would be useful to add noise such that we define a valid spread divergence, but can maximally still discern the difference between the two distributions. To gain intuition, we define $p$ and $q$ to generate data in separated linear subspaces, $p(x) = \int \delta\left(x - a - Az\right)p(z)dz$, $q(x) = \int \delta\left(x - b - Bz\right)p(z)dz$, $p(z) = \mathcal{N}\left(z|0, I_Z\right)$. Using Gaussian spread, $p(y|x) = \mathcal{N}\left(y|\mu, \Sigma\right)$, what is the optimal $\mu, \Sigma$ that maximises the divergence? Clearly, as $\Sigma$ tends to zero, the divergence increases to infinity, meaning that we must at least constrain the entropy of $p(y|x)$ to be finite. In this case the spreaded distributions are given by

$$\tilde{p}(y) = \mathcal{N}\left(y|\mu + a, AA^\mathsf{T} + \Sigma\right), \qquad \tilde{q}(y) = \mathcal{N}\left(y|\mu + b, BB^\mathsf{T} + \Sigma\right) \quad (16)$$

We define a simple Factor Analysis noise model with $\Sigma = \sigma^2 I + uu^\mathsf{T}$, where $\sigma^2$ is fixed and $u^\mathsf{T}u = 1$. The entropy of $p(y|x)$ is then fixed and independent of $u$. Also, for simplicity, we assume $A = B$. It is straightforward to show that the spread divergence $\mathrm{KL}(\tilde{p}||\tilde{q})$ is maximised for $u$ pointing orthogonal to the vector $\left(AA^\mathsf{T} + \sigma^2 I\right)^{-1}(b - a)$. Then $u$ optimally points along the direction in which the support lies. The support of $p(y|x)$ must be the whole space but to maximise the divergence the noise preferentially spreads along directions defined by $p$ and $q$, see figure(1).

---

[2] If $\mathcal{F}\{K\}$ can change sign, by continuity, there must exist a point at which $\mathcal{F}\{K\} = 0$.

[3] This includes the Laplace, Mellin and Hartley transforms.

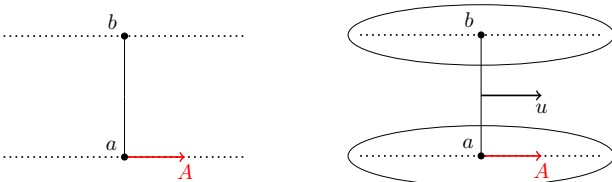

Figure 1: Left: The lower dotted line denotes Gaussian distributed data $p(x)$ with support only along the linear subspace defined by the origin $a$ and direction $A$. The upper dotted line denotes Gaussian distributed data $q(x)$ with support different from $p(x)$. Optimally, to maximise the spread divergence between the two distributions, for fixed noise entropy, we should add noise that preferentially spreads out along the directions defined by $p$ and $q$, as denoted by the ellipses.

## 4 MERCER SPREAD DIVERGENCE

We showed in section(3) how to define one form of spread divergence, with the result that stationary noise distributions must have strictly positive Fourier Transforms. A natural question is whether, for continuous $x$, there are other easily definable noise distributions that are non-stationary. To examine this question, let $x \in [a, b]$, $y \in [a, b]$ and $K(x, y) = K(y, x)$ be square integrable, $K(x, y) \in L^2$. We define Mercer noise $p(y|x) = K(x, y)/K(x)$, where $K(x) = \int K(x, y) dy$. For strictly positive definite $K$, by Mercer's Theorem, it admits an expansion

$$K(x, y) = \sum_n \lambda_n \phi_n(x) \phi_n(y) \tag{17}$$

where the eigenfunctions $\phi_n$ form a complete orthogonal set of $L^2[a, b]$ and all $\lambda_n > 0$, see for example Sriperumbudur et al. (2011). Then

$$\tilde{p}(y) = \sum_n \int \lambda_n \phi_n(x) \phi_n(y) \frac{p(x)}{K(x)} dx, \qquad \tilde{q}(y) = \sum_n \int \lambda_n \phi_n(x) \phi_n(y) \frac{q(x)}{K(x)} dx \tag{18}$$

and $\tilde{p}(y) = \tilde{q}(y)$ is equivalent to the requirement

$$\sum_n \int \lambda_n \phi_n(x) \phi_n(y) \frac{p(x)}{K(x)} dx = \sum_n \int \lambda_n \phi_n(x) \phi_n(y) \frac{q(x)}{K(x)} dx \tag{19}$$

Multiplying both sides by $\phi_m(y)$ and integrating over $y$ we obtain

$$\int \phi_m(x) \frac{p(x)}{K(x)} dx = \int \phi_m(x) \frac{q(x)}{K(x)} dx \tag{20}$$

If $p(x)/K(x)$ and $q(x)/K(x)$ are in $L^2[a, b]$ then, from Mercer's Theorem, they can be expressed as orthogonal expansions

$$\frac{p(x)}{K(x)} = \sum_n \gamma_n^p \phi_n(x), \qquad \frac{q(x)}{K(x)} = \sum_n \gamma_n^q \phi_n(x) \tag{21}$$

Then, equation(20) is

$$\int \phi_m(x) \sum_n \gamma_n^p \phi_n(x) dx = \int \phi_m(x) \sum_n \gamma_n^q \phi_n(x) dx \tag{22}$$

which reduces to (using orthonormality), $\gamma_m^p = \gamma_m^q \Rightarrow p = q$. Hence, provided $K(x, y) = K(y, x)$ is square integrable on $[a, b]$ and strictly positive definite, then $K(x, y)/\int K(x, y) dy$ defines valid spread noise. For example, $K(x, y) = \exp\left(-\lambda_1(x^2 + y^2)\right) + \exp\left(-\lambda_2(x^2 + y^2)\right)$ defines a strictly positive non-stationary square integrable kernel on $[a, b]$. Provided $p(x)/K(x)$ and $q(x)/K(x)$ are in $L^2[a, b]$ then the spread noise $p(y|x) = K(x, y)/K(x)$ defines a valid spread divergence.

## 5   Applications

We demonstrate using a spread divergence to train implicit models

$$p_\theta(x) = \int \delta\left(x - g_\theta(z)\right) p(z) dz \tag{23}$$

where $\theta$ are the parameters of the encoder $g$. We show that, despite the likelihood not being defined, we can nevertheless successfully train the models using an EM style algorithm, see for example Barber (2012). We then show how to train a deterministic non-linear generative model using a variational approximation.

### 5.1   Deterministic Linear Latent Model

For observation noise $\gamma$, the Probabilistic PCA model (Tipping & Bishop, 1999) for $X$-dimensional observations and $Z$-dimensional latent is

$$x = Fz + \gamma\epsilon, \quad z \sim N(0, I_Z), \quad \epsilon \sim N(0, I_X), \quad p_\theta(x) = \mathcal{N}\left(y|0, FF^\mathsf{T} + \gamma^2 I_X\right) \tag{24}$$

When $\gamma = 0$, the generative mapping from $z$ to $x$ is deterministic and the model $p_\theta(x)$ has support only on a subset of $\mathbb{R}^X$ and the data likelihood is in general not defined. In the following we consider general $\gamma$, setting $\gamma$ to zero at the end of the calculation. To fit the model to iid data $\{x_1, \ldots, x_N\}$ using maximum likelihood, the only information required from the dataset is the data covariance $\hat{\Sigma}$. The maximum likeihood solution for PPCA is then $F = U_Z \left(\Lambda_Z - \gamma^2 I_Z\right)^{\frac{1}{2}} R$, where $\Lambda_Z$, $U_Z$ are the $Z$ largest eigenvalues, eigenvectors of $\hat{\Sigma}$; $R$ is an arbitrary orthogonal matrix. Using spread noise $p(y|x) = \mathcal{N}\left(y|x, \sigma^2 I_X\right)$, the spreaded distribution $\tilde{p}_\theta(y)$ is a Gaussian

$$\tilde{p}_\theta(y) = \mathcal{N}\left(y|0, FF^\mathsf{T} + (\gamma^2 + \sigma^2)I_X\right) \tag{25}$$

Thus, $\tilde{p}_\theta(y)$ is of the same form as PPCA, albeit with an inflated covariance matrix. Adding Gaussian spread noise to the data also simply inflates the sample covariance to $\hat{\Sigma}' = \hat{\Sigma} + \sigma^2 I_X$. Since the eigenvalues of $\hat{\Sigma}' \equiv \hat{\Sigma} + \sigma^2 I_X$ are simply $\Lambda' = \Lambda + \sigma^2 I_X$, with unchanged eigenvectors, the optimal deterministic ($\gamma = 0$) latent linear model has solution $F = U_Z \left(\Lambda'_Z - \sigma^2 I_Z\right)^{\frac{1}{2}} R = U_Z \Lambda_Z^{\frac{1}{2}} R$. Unsurprisingly, this is the standard PCA solution; however, the derivation is non-standard since the likelihood of the deterministic latent linear model is not defined. Nevertheless, using the spread divergence, we learn a sensible model and recover the true data generating process if the data were exactly generated according to the deterministic model.

### 5.2   Deterministic Independent Components Analysis

ICA corresponds to the model

$$p(x, z) = p(x|z) \prod_i p(z_i) \tag{26}$$

where the independent components $z_i$ follow a non-Gaussian distribution. For Gaussian noise ICA an observation $x$ is assumed to be generated by the process

$$p(x|z) = \prod_j \mathcal{N}\left(x_j | g_j(z), \gamma^2\right) \tag{27}$$

where $g_i(z)$ mixes the independent latent process $z$. In standard linear ICA, $g_j(z) = a_j^\mathsf{T} z$ where $a_j$ is the $j^{th}$ column on the mixing matrix $A$. For small observation noise $\gamma^2$, the EM algorithm (Bermond & Cardoso, 1999) becomes ineffective. To see this, consider $X = Z$ and invertible mixing matrix $A$, $x = Az$. At iteration $k$ the EM algorithm has an estimate $A_k$ of the mixing matrix. The M-step updates $A_k$ to

$$A_{k+1} = \mathbb{E}\left[xz^\mathsf{T}\right] \mathbb{E}\left[zz^\mathsf{T}\right]^{-1} \tag{28}$$

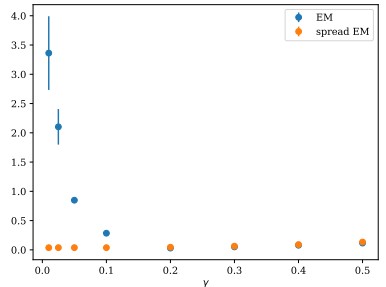

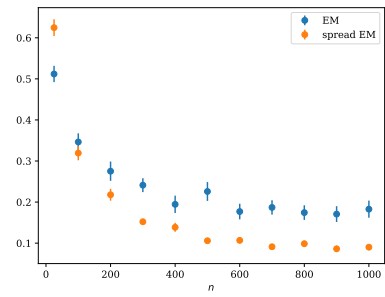

(a) Relative error $|A_{ij}^{est} - A_{ij}^{true}|/|A_{ij}^{true}|$ as a function of the model noise standard deviation $\gamma$.

(b) Relative error $|A_{ij}^{est} - A_{ij}^{true}|/|A_{ij}^{true}|$ as a function of the number of datapoints $N$.

Figure 2: (a) For $X = 20$ observations and $Z = 10$ latent variables, we generate $N = 20000$ datapoints from the model $x = Az$, for independent zero mean unit variance Laplace components on $z$. The elements of $A$ used to generate the data are uniform random $\pm 1$. We use $S_y = 1$, $S_z = 1000$ samples and 2000 EM iterations to estimate the mixing matrix. The relative error is averaged over all $i, j$ and 10 random experiments. We also plot standard errors around the mean relative error. In blue we show the error in learning the underlying parameter using the standard EM algorithm. As expected, as $\gamma \to 0$, the error blows up as the EM algorithm 'freezes'. In orange we plot the error for EM using spread noise, as described in section(5.2.1); no slowing down appears as the model noise $\gamma$ decreases. As the model noise increases, the quality of the learned model under spread noise decreases gradually. In (b) we show that, apart from very small $N$, the error for the spread EM algorithm is lower than for the standard EM algorithm. Here $Z = 5$, $X = 10$, $S = 1000$, $\gamma = 0.2$, with 500 EM updates used. Results are averaged over 50 runs of randomly drawn $A$.

where, for noiseless data ($\gamma = 0$),

$$\mathbb{E}\left[xz^\mathsf{T}\right] = \frac{1}{N}\sum_n x_n \left(A_k^{-1}x_n^\mathsf{T}\right) = \hat{S}A_k^{-\mathsf{T}}, \quad \mathbb{E}\left[zz^\mathsf{T}\right] = A_k^{-1}\hat{S}A_k^{-\mathsf{T}} \tag{29}$$

where $\hat{S} \equiv \frac{1}{N}\sum_n x_n x_n^\mathsf{T}$ is the moment matrix of the data. Thus, $A_{k+1} = \hat{S}A_k^{-\mathsf{T}}\left(A_k^{-1}\hat{S}A_k^{-\mathsf{T}}\right)^{-1} = A_k$. and the algorithm 'freezes'. Similarly, for low noise $\gamma \ll 1$ progress critically slows down. Whilst over-relaxation methods, see for example Winther & Petersen (2007) can help in the case of small noise, for zero noise $\gamma = 0$, over-relaxation is of no benefit.

### 5.2.1 Healing Critical Slowing Down

To deal with small noise and the limiting case of a deterministic model ($\gamma = 0$), we consider Gaussian spread noise $p(y|x) = \mathcal{N}\left(y|x, \sigma^2 I_X\right)$ to give

$$p(y, z) = \int p(y|x)p(x, z)dx = \prod_i \mathcal{N}\left(y|g_i(z), \left(\gamma^2 + \sigma^2\right)I_X\right)\prod_i p(z_i) \tag{30}$$

The empirical distribution is replaced by the spreaded empirical distribution

$$\hat{p}(y) = \frac{1}{N}\sum_n \mathcal{N}\left(y|x^n, \sigma^2 I_X\right) \tag{31}$$

The M-step has the same form as equation(28) but with modified statistics

$$\mathbb{E}\left[yz^\mathsf{T}\right] = \frac{1}{N}\sum_n \int \mathcal{N}\left(y|x^n, \sigma^2\right)p(z|y)yz^\mathsf{T}dzdy, \tag{32}$$

$$\mathbb{E}\left[zz^\mathsf{T}\right] = \frac{1}{N}\sum_n \int \mathcal{N}\left(y|x^n, \sigma^2\right)p(z|y)zz^\mathsf{T}dzdy \tag{33}$$

The E-step optimally sets

$$p(z|y) = \frac{1}{Z_q(y)}\mathcal{N}\left(z|\mu(y), \Sigma\right)\prod_i p(z_i), \quad Z_q(y) = \int \mathcal{N}\left(z|\mu(y), \Sigma\right)\prod_i p(z_i)dz \tag{34}$$

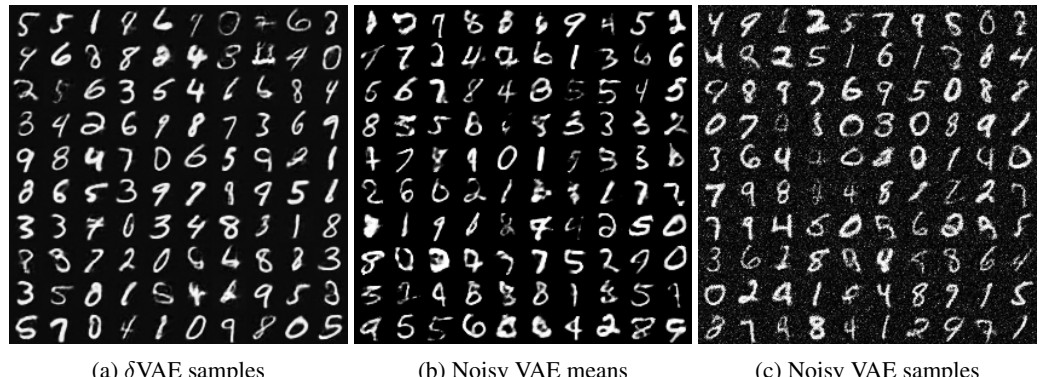

(a) $\delta$VAE samples    (b) Noisy VAE means    (c) Noisy VAE samples

Figure 3: Comparison of deep generative models trained on the MNIST digits after 300K iterations. (a) Samples from the trained deterministic model. (b) Means from a standard (noisy) VAE with fixed observation noise. (c) Samples from the standard noisy VAE model.

where $Z_q(y)$ is a normaliser and

$$\Sigma = (\gamma^2 + \sigma^2)\left(A^\mathsf{T}A\right)^{-1}, \quad \mu(y) = \left(A^\mathsf{T}A\right)^{-1}Ay \tag{35}$$

We can rewrite the expectations required for the E-step of the EM algorithm as

$$\mathbb{E}\left[yz^\mathsf{T}\right] = \frac{1}{N}\sum_n \int \mathcal{N}\left(y|x^n, \sigma^2\right)\mathcal{N}\left(z|\mu(y), \Sigma\right)\frac{\prod_i p(z_i)}{Z_q(y)}yz^\mathsf{T}dzdy \tag{36}$$

$$\mathbb{E}\left[zz^\mathsf{T}\right] = \frac{1}{N}\sum_n \int \mathcal{N}\left(y|x^n, \sigma^2\right)\mathcal{N}\left(z|\mu(y), \Sigma\right)\frac{\prod_i p(z_i)}{Z_q(y)}zz^\mathsf{T}dzdy \tag{37}$$

Generally the posterior $p(z|y)$ will be peaked around $\mathcal{N}\left(z|\mu(y), \Sigma\right)$ and writing the expectations with respect to $\mathcal{N}\left(z|\mu(y), \Sigma\right)$ allows for an effective sampling approximation focussed on regions of high probability. We implement this update by drawing $S_y$ samples from $\mathcal{N}\left(y|x_n, \sigma^2 I\right)$ and, for each $y$ sample, we draw $S_z$ samples from $\mathcal{N}\left(z|\mu(y), \Sigma\right)$. This scheme has the advantage over more standard variational approaches, see for example Winther & Petersen (2007), in that we obtain a consistent estimator of the M-step update for $A$[4]. We show results for a toy experiment in figure(2), learning the underlying mixing matrix in a deterministic non-square setting. Note that standard algorithms such as FastICA (Hyvärinen, 1999) fail in this setting. The noise value is set to $\sigma = \max(0.001, 2.5 * \text{sqrt}(\text{mean}(AA^\mathsf{T})))$, for estimated mixing matrix $A$ of the underlying deterministic model $x_n = Az_n$, $n = 1, \ldots, N$. The EM algorithm learns a good approximation of the unknown mixing matrix and latent components $z_n$, with no critical slowing down.

### 5.3 Training Implicit Non-linear Models

For a deterministic non-linear implicit model, we set $p(z) = \mathcal{N}\left(z|0, I\right)$ and parameterise $g_\theta(x)$ by a deep neural network. The likelihood equation(23) is in general intractable and it is natural to consider a variational approximation (Kingma & Welling, 2013),

$$\log p_\theta(x) \geq -\int q_\phi(z|x)\left(\log q_\phi(z|x) + \log\left(p_\theta(x|z)p(z)\right)\right)dz \tag{38}$$

However, since $p_\theta(x|z) = \delta\left(x - g_\theta(z)\right)$ this bound is not well defined. Instead, we minimise the spread divergence $\text{KL}(\tilde{p}(y)||\tilde{p}_\theta(y))$. The approach is a straightforward extension of the standard

---

[4]We focus on demonstrating how the spread divergences heals critical slowing down, rather than deriving a state-of-the-art approximation of $p(z|y)$. The importance sampling approach has fast run time and works well, even for large latent dimensions, $Z = 50$. We also implemented a variational factorised approximation of $p(z|y)$ but found this to be relatively slow and ineffective. A variational Gaussian approximation of $p(z|y)$ improves on the factorised approximation, but is still slow compared to the importance sampling scheme.

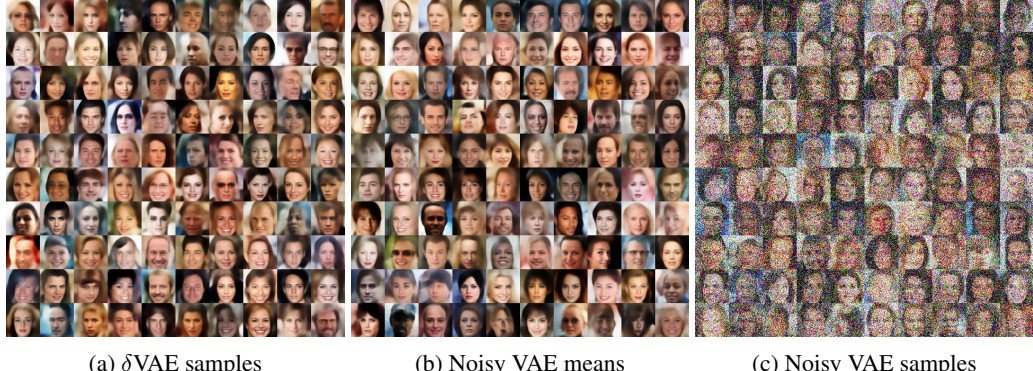

| (a) $\delta$VAE samples | (b) Noisy VAE means | (c) Noisy VAE samples |

Figure 4: Comparison of training approaches for the CelebA dataset. All models had the same structure and were trained using the same Adam settings, as in the MNIST experiment.

variational autoencoder and in appendix(C) we provide details of how to do this, along with higher resolution images of samples from the generative model. We dub this model and associated spread divergence training the '$\delta$VAE'. As a demonstration, we trained a generative network on the MNIST dataset, see figure(3) and appendix(D). We used Gaussian spread noise $\sigma = 1$ for the $\delta$VAE and observation noise $\sigma = 0.5$ for the standard noisy VAE. The network $g_\theta(x)$ contains 8 layers, each layer with 400 units and relu activation function and latent dimension $Z = 64$. We also trained a deep convolutional generative model on the CelebA dataset (Liu et al., 2015), see figure(4) and appendix(E). We pre-process CelebA images by first taking 140x140 centre crops and then resizing to 64x64. Pixel values were then rescaled to lie in $[0, 1]$. We use Gaussian spread noise $\sigma = 0.5$ for the $\delta$VAE and observation noise $\sigma = 0.5$ for the standard noisy VAE.

## 6 SUMMARY

We described an approach to defining a divergence, even when two distributions to not have the same support. The method introduces a 'noise' variable to 'spread' mass from each distribution to cover the same domain. Previous approaches (Furmston & Barber, 2009; Sønderby et al., 2016) can be seen as special cases. We showed that defining divergences this way enables us to train deterministic generative models using standard 'likelihood' based approaches. Indeed, for simple models such as Independent Components Analysis, we showed how we can implement a principled learning method based on classical EM training, without the standard difficulty of critical slowing down in the case of small (or zero) observation noise.

Introducing noise means that an additional expectation is required. This can be carried out, in part, exactly, although additional approximations using perturbation theory are possible, similar to Roth et al. (2017). Spread divergences have deep connections to other approaches to define measures of disagreement between distributions. In particular, one can view the spread divergence as the probabilistic analogue of MMD, with conditions required for the existence of the spread divergence closely related to the universality requirement on MMD kernels (Micchelli et al., 2006).

Theoretically, we can learn the underlying true data generating process by the use of any valid spread divergence — for example for fixed Gaussian spread noise. In practice, however, the quality of the learned model can depend on the choice of spread noise. In this work we fixed the spread noise, but showed that if we were to learn the spread noise, it would preferentially spread mass across the manifolds defining the two distributions. In future work, we will investigate learning spread noise to maximally discriminate two distributions, which would involve a minimax model training objective, with an inner maximisation over the spread noise and an outer maximisation over the model parameters. This would bring our work much closer to adversarial training methods (Goodfellow, 2017).

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

## A  SPREAD NOISE MAKES DISTRIBUTIONS MORE SIMILAR

The data processing inequality for $f$-divergences (see for example Gerchinovitz et al. (2018)) states that $D_f(\tilde{p}(y)||\tilde{q}(y)) \leq D_f(p(x)||q(x))$. For completeness, we provide here an elementary proof of this result. We consider the following joint distributions

$$q(y,x) = p(y|x)q(x), \qquad p(y,x) = p(y|x)p(x) \tag{39}$$

whose marginals are the spreaded distributions

$$\tilde{p}(y) = \int_x p(y|x)p(x), \qquad \tilde{q}(y) = \int_x p(y|x)q(x) \tag{40}$$

The divergence between the two joint distributions is

$$D_f(p(y,x)||q(y,x)) = \int_{x,y} q(y,x)f\left(\frac{p(y|x)p(x)}{p(y|x)q(x)}\right) = D_f(p(x)||q(x)) \tag{41}$$

The $f$-divergence between two marginal distributions is no larger than the $f$-divergence between the joint (see also Zhang et al. (2018)). To see this, consider

$$\begin{aligned}
D_f(p(u,v)||q(u,v)) &= \int q(u) \int q(v|u)f\left(\frac{p(u,v)}{q(u,v)}\right) dydu \\
&\geq \int q(u)f\left(\int q(v|u)\frac{p(u,v)}{q(v|u)q(u)}dv\right) du \\
&= \int q(u)f\left(\frac{p(u)}{q(u)}\right) du = D_f(p(u)||q(u))
\end{aligned}$$

Hence,

$$D_f(\tilde{p}(y)||\tilde{q}(y)) \leq D_f(p(y,x)||q(y,x)) = D_f(p(x)||q(x)) \tag{42}$$

Intuitively, spreading two distributions increases their overlap, reducing the divergence. When $p$ and $q$ do not have the same support, $D_f(q(x)||p(x))$ can be infinite or not well-defined.

## B  INJECTIVE LINEAR MAPPINGS

Consider an injective linear mapping $T$ from space $V$ to $W$. From the rank nullity theorem for finite dimensional spaces,

$$\dim(\text{image}(T)) + \dim(\text{kernel}(T)) = \dim(V) \tag{43}$$

If $T$ is injective, then $\dim(\text{kernel}(T)) = 0$. If $\dim(V) = \dim(W)$ then

$$\dim(\text{image}(T)) = \dim(W) \tag{44}$$

Since $\text{image}(T) \subseteq W$, it must be that $\text{image}(T) = W$. Hence, injective linear maps between between two (finite dimensional) spaces of the same dimension are surjective; equivalently, they are invertible.

In the context of spread noise, since the domain of $x$ and $y$ are equal and $\tilde{p}(y)$ is defined through a linear transformation of $p(x)$, the requirement in (5) that the mapping is injective is equivalent to the requirement that the mapping is invertible.

## C  SPREAD DIVERGENCE FOR DETERMINISTIC DEEP GENERATIVE MODELS

Instead of minimising the likelihood, we train an implicit generative model by minimising the spread divergence

$$\min_\theta \text{KL}(\tilde{p}(y)||\tilde{p}_\theta(y)) \tag{45}$$

where

$$\tilde{p}(y) = \frac{1}{N} \sum_{n=1}^{N} \mathcal{N}\left(y|x_n, \sigma^2 I_X\right) \tag{46}$$

and

$$\tilde{p}_\theta(y) = \int p(y|x)p_\theta(x)dx = \int \mathcal{N}\left(y|g_\theta(z), \sigma^2 I_X\right) p(z)dz = \int p_\theta(y|z)p(z)dz \tag{47}$$

According to our general theory,

$$\min_\theta \mathrm{KL}(\tilde{p}(y)||\tilde{p}_\theta(y)) = 0 \quad \Leftrightarrow \quad p(x) = p_\theta(x) \tag{48}$$

Here

$$\mathrm{KL}(\tilde{p}(y)||\tilde{p}_\theta(y)) = \frac{1}{N} \sum_{n=1}^{N} \int \mathcal{N}\left(y|x_n, \sigma^2 I_X\right) \log \tilde{q}(y)dy + const. \tag{49}$$

Typically, the integral over $y$ will be intractable and we resort to an unbiased sampled estimate (though see below for Gaussian $q$). Neglecting constants, the KL divergence estimator is

$$\frac{1}{NS} \sum_{n=1}^{N} \sum_{s=1}^{S} \log \tilde{q}(y_s^n) \tag{50}$$

where $y_s^n$ is a noisy sample of $x_n$, namely $y_s^n \sim \mathcal{N}\left(y_s^n|x_n, \sigma^2 I_X\right)$. In most cases of interest, with non-linear $g$, the distribution $\tilde{q}(y)$ is intractable. We therefore use the variational lower bound

$$\log \tilde{p}_\theta(y) \geq \int q_\phi(z|y) \left(- \log q_\phi(z|y) \log \left(p_\theta(y \mid z)p(z)\right)\right) dz \tag{51}$$

Parameterising the variational distribution as a Gaussian,

$$q_\phi(z|y) = \mathcal{N}\left(z|\mu_\phi(y), \Sigma_\phi(y)\right) \tag{52}$$

then we can reparameterise and write

$$\log \tilde{p}_\theta(y) \geq H(\Sigma_\phi) + \mathbb{E}_{\mathcal{N}(\epsilon|0,I)} \left[\log \left(p_\theta(y|z = \mu_\phi + C_\phi\epsilon)q_z(z = \mu_\phi + C_\phi\epsilon)\right)\right] \tag{53}$$

where $H$ is the entropy of a Gaussian with covariance $\Sigma_\phi$. For Gaussian spread noise in $D$ dimensions, this is (ignoring constants)

$$\log \tilde{p}_\theta(y) \geq H(\Sigma_\phi) + \mathbb{E}_{\mathcal{N}(\epsilon|0,I)} \left[-\frac{1}{(2\sigma^2)^{D/2}} \left(y - g_\theta\left(\mu_\phi(y) + C_\phi\epsilon\right)\right)^2 + \log q_z(z = \mu_\phi(y) + C_\phi\epsilon)\right] \tag{54}$$

where $C_\phi$ is the Cholesky decomposition of $\Sigma_\phi$.

The overall procedure is therefore a straightforward modification of the standard VAE method Kingma & Welling (2013) in which both the model and data are corrupted by noise:

1. Choose a noise corruption variance $\sigma^2$.

2. Choose a tractable family for the variational distribution, for example $q_\phi(z|y) = \mathcal{N}\left(z|\mu_\phi(y), \Sigma_\phi(y)\right)$ and initialise $\phi$.

3. We then sample a noisy version $y_n$ for each datapoint (if we're using $S = 1$ samples)

4. Draw samples $\epsilon$ to estimate $\log \tilde{p}_\theta(y_n)$, equation(54)

5. Do a gradient ascent step in $(\theta, \phi)$.

6. Go to 3 and repeat until convergence.

We note that for $\Sigma_\phi$ independent of $y$, we can partially integrate equation(54) over $y$ to give the bound

$$\int \mathcal{N}\left(y|x,\sigma^2 I_X\right) \log \tilde{p}_\theta(y) \geq H(\Sigma_\phi) + \mathbb{E}_{\mathcal{N}(\epsilon|0,I)}\left[\log q_z(z = \mu_\phi(y) + C_\phi \epsilon)\right] \tag{55}$$

$$- \frac{1}{(2\sigma^2)^{D/2}} \mathbb{E}_{\mathcal{N}(\epsilon|0,I)}\left[\mathbb{E}_{\mathcal{N}(y|x,\sigma^2 I_X)}\left[(y - g_\theta\left(\mu_\phi(y) + C_\phi \epsilon\right))^2\right]\right] \tag{56}$$

where

$$\mathbb{E}_{\mathcal{N}(y|x,\sigma^2 I_X)}\left[(y - g_\theta\left(\mu_\phi(y) + C_\phi \epsilon\right))^2\right]$$
$$= \sigma^2 - 2\mathbb{E}_{\mathcal{N}(\epsilon_x|0,I_X)}\left[\epsilon_x g_\theta(\mu_\phi(x + \sigma\epsilon_x))\right] + \mathbb{E}_{\mathcal{N}(\epsilon_x|0,I_X)}\left[(x - g_\theta(\mu_\phi\left(x + \sigma\epsilon_x\right)))^2\right] \tag{57}$$

Similar to Roth et al. (2017), in principle, one can form a perturbation approximation of the above to second order in $\epsilon_x$ and express the integral over the spread noise as a form of regularisation; however, in our experiments we found that the above works well – we therefore leave such analysis for future work.

## D  MNIST EXPERIMENT

We first scaled the MNIST data to lie in $[0.05, 0.95]$ and then transformed using the logit (inverse logistic sigmoid) of the pixel value in order to use Gaussian spread noise. We use Gaussian spread noise $\sigma = 1$ for the $\delta$VAE and observation noise $\sigma = 0.5$ for the standard noisy VAE. The network $g_\theta(x)$ contains 8 layers, each layer with 400 units and relu activation function and latent dimension $Z = 64$. The variational inference network $q_\phi(z|y) = \mathcal{N}\left(z|\mu_\phi(y), \sigma_\phi^2 I_Z\right)$ has a similar structure for the mean network $\mu_\phi(y)$. Learning was done using the Adam optimiser with learning rate $10^{-4}$ and exponential decay rate of 0.96 every 10000 iterations.

## E  CELEBA EXPERIMENT

Both encoder and decoder used fully convolutional architectures with 5x5 convolutional filters and used vertical and horizontal strides 2 except the last deconvolution layer we used stride 1. Here $\text{Conv}_k$ stands for a convolution with $k$ filters, $\text{DeConv}_k$ for a deconvolution with k filters, BN for the batch normalization Ioffe & Szegedy (2015), ReLU for the rectified linear units, and $\text{FC}_k$ for the fully connected layer mapping to $R^k$.

$$x \in R^{64 \times 64 \times 3} \rightarrow \text{Conv}_{128} \rightarrow \text{BN} \rightarrow \text{Relu}$$
$$\rightarrow \text{Conv}_{256} \rightarrow \text{BN} \rightarrow \text{Relu}$$
$$\rightarrow \text{Conv}_{512} \rightarrow \text{BN} \rightarrow \text{Relu}$$
$$\rightarrow \text{Conv}_{1024} \rightarrow \text{BN} \rightarrow \text{Relu} \rightarrow \text{FC}_{64}$$

$$z \in R^{64} \rightarrow \text{FC}_{8 \times 8 \times 1024}$$
$$\rightarrow \text{DeConv}_{512} \rightarrow \text{BN} \rightarrow \text{Relu}$$
$$\rightarrow \text{DeConv}_{256} \rightarrow \text{BN} \rightarrow \text{Relu}$$
$$\rightarrow \text{DeConv}_{128} \rightarrow \text{BN} \rightarrow \text{Relu}$$
$$\rightarrow \text{DeConv}_{64} \rightarrow \text{BN} \rightarrow \text{Relu} \rightarrow \text{DeConv}_3$$

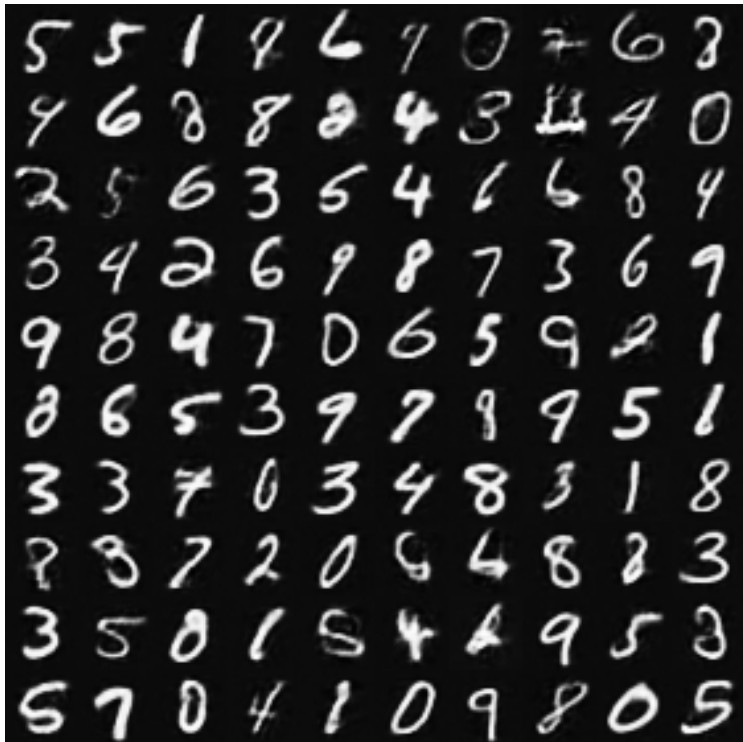

Figure 5: $\delta$VAE samples

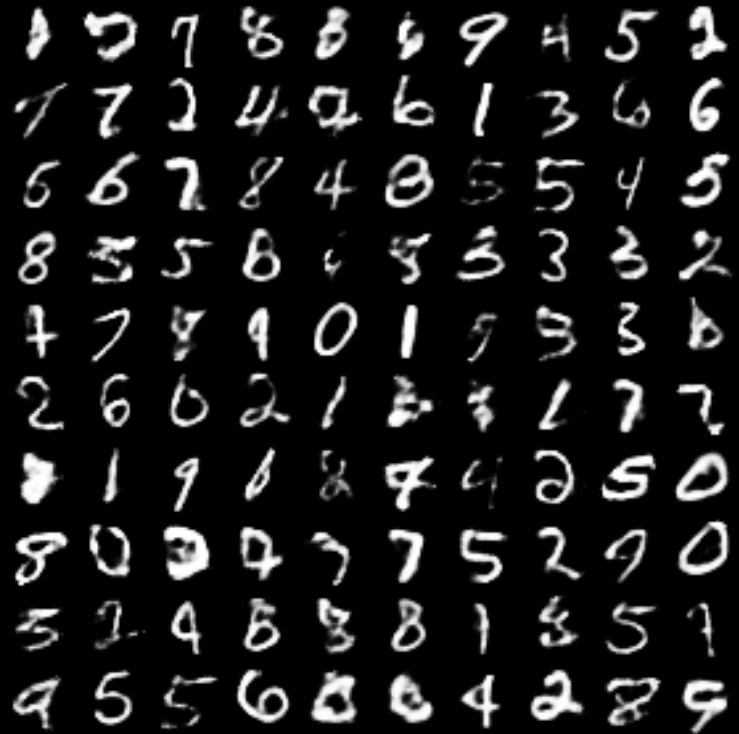

Figure 6: Noisy VAE means

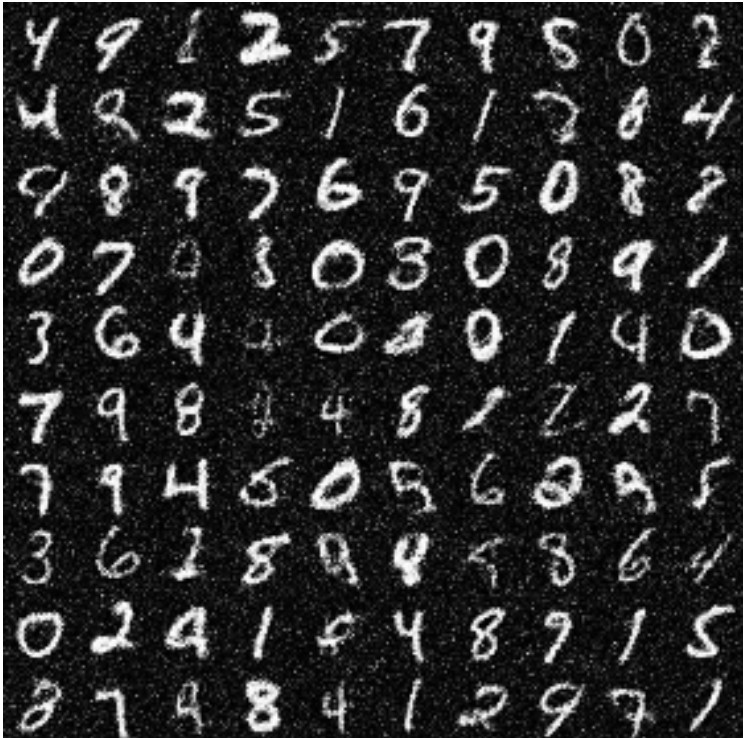

Figure 7: Noisy VAE samples

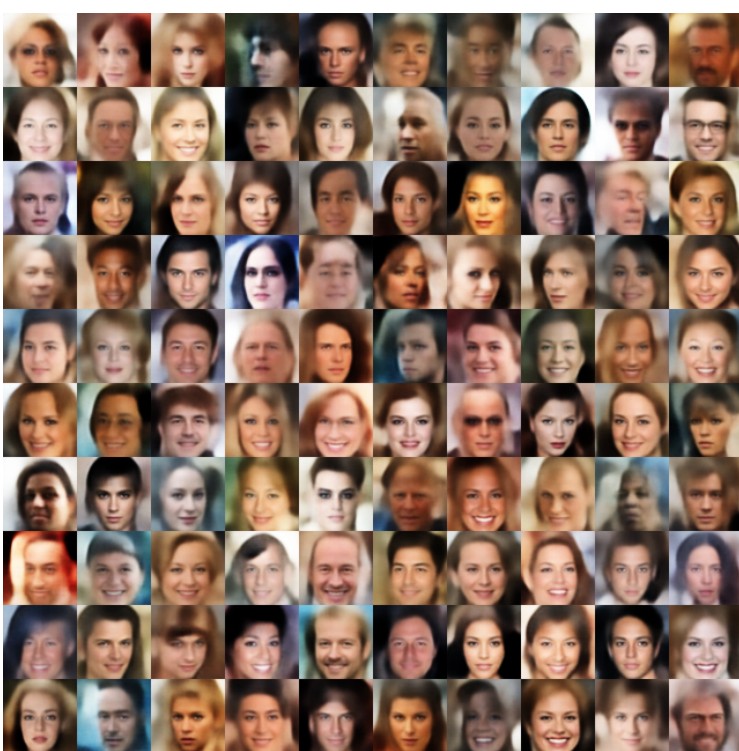

Figure 8: $\delta$VAE samples

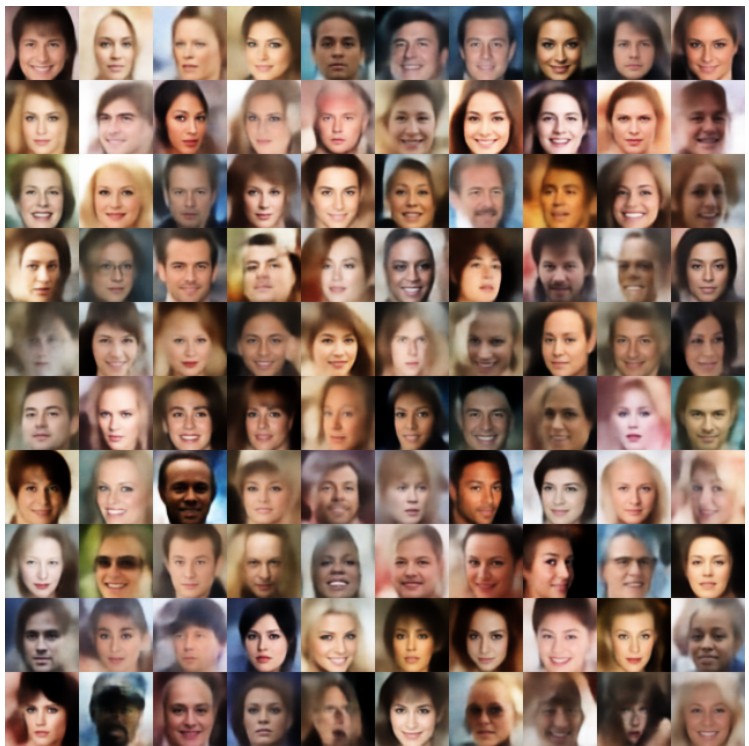

Figure 9: Noisy VAE means

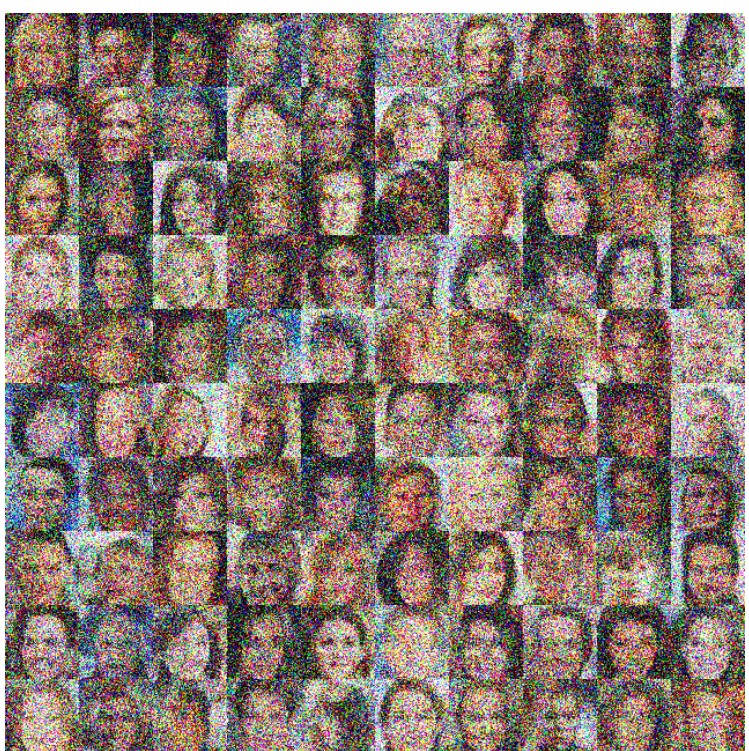

Figure 10: Noisy VAE samples

