# OpenReview forum: "Spread Divergences"
_ICLR.cc/2019/Conference_

### Official Review · AnonReviewer2 · 2018-11-01
**Good theoretical contribution; empirical results could be more extensive**

**Rating:** 6
**Confidence:** 4

**Review:**

Summary
=======
This paper introduces spread divergences. Spread divergences are obtained by taking the divergence between smoothed/noisy versions of two distributions. A spread divergence between two distributions of non-overlapping support can be defined even when the corresponding divergence is not. The authors discuss conditions under which the data generating process can be identified by minimizing spread divergences and apply spread divergences to the examples of PCA, ICA, and noiseless VAE.

Review
======
With a lot of papers focusing on generative modeling, divergence minimization is of great relevance to the ICLR community. Adding noise to distributions to ensure overlapping support is intuitive and has been used to stabilize training of GANs, but I am unaware of any work focusing on questions of identifiability and efficiency. I especially like the example of slowing EM in ICA with small noise. Here, some empirical results are lacking which analyze the speed/correctness of the identification of parameters for various choices of divergence/model noise. These would have greatly enhanced the paper. Instead, the available space was used to show model samples, which I find less helpful.

In Section 3.2 and Section 6 the authors argue that choosing noise which maximizes the spread divergence is optimal or at least preferable. This seems counterintuitive, given that the point of the noise was to make the distributions more similar. Please elaborate on why maximizing the divergence is a good strategy.

Minor
=====
The paper seems hastily written, with some grammar issues, typos, and sloppy use of LaTeX, e.g.:

– "-\log" instead of "\log" in definition of KL divergence in the introduction
– "Section 2.2" not "section(2.2)", "Equation 24" not "equation(24)"
– "model (Tipping & Bishop, 1999)" instead of "model Tipping & Bishop (1999)"
– "\mid" instead of "|"
– "x" instead of "y" in Equation 28

Please provide a reference for the EM algorithm of ICA.

---

> ### Author Response · Authors · 2018-11-22
> **response**
>
> * We agree that the proposed experiments will be useful and we will include them.
>
> * The intuition for maxmising the divergence is similar to that in MMD and Wasserstein distance, namely that we wish to consider mappings which maximally enable us to discern the difference between two distrubutions (whilst retaining that the distance/divergence is well defined). We will update the text to try to make this point more clear.]
>
> * Thanks for the minor details suggestions -- we'll update, though feel that section and equation citing are stylistic preferences, and not errors as such.
>
> * We will add a reference to EM and ICA.

---

### Official Review · AnonReviewer3 · 2018-11-01
**A narrow focused paper for an interesting idea, forgetting the SOTA for the formal part, whose shape could be much improved**

**Rating:** 4
**Confidence:** 4

**Review:**


Pros:

- interesting idea.

Cons:

- the paper forgets the state of the art for comparisons (optimal transport, data processing inequalities)
- the paper formally shows little as most results are in fact buried in the text and it is hard to tell the formal from the informal.
- experiments fall short of really using the setting proposed
- the paper focuses too much on keeping the identity of the indiscernibles and forgets the study of other properties (including downsides, such as variance increase)

Detail:

* The paper claims to propose a "theory" for spread divergences (conditioning a f-divergence by a "third-party" conditional distribution on supports which makes supports match) still keeping the identity of indiscernibles.

* The paper recycles the notion of Spread f-divergences from Zhang et al. (which makes a circular reference to this paper for the introduction of these divergences).

* The paper motivates the notion of spread divergences by the fact that f-divergences impose matching supports (Section 1), not mentioning that optimal transport theory is a much natural fit for any such kind of setting (e.g. Wasserstein distances). This is a big omission and a missed occasion for a potentially interesting discussion.

* The paper then claims that "spread noise makes distributions more similar" (Section 2.1), not mentioning that equation (8), which it claims to have been shown by Zhang et al. paper (see below), is in fact a data processing inequality *long known*. They will find it, along with a huge number of other useful properties, in series of IEEE T. IT papers, among which Pardo and Vajda's "About distances of discrete distributions satisfying the data processing theorem of information theory",  Van Erven and Harremoes, "Re ́nyi Divergence and Kullback-Leibler Divergence" (for the KL / Rényi divergence, but you have more references inside), etc. .

* The paper then goes on "showing" (a word used often, even when there is not a single Theorem, Lemma or the like ever stated in the paper...) several properties (Section 2.2). The first states that (9) is equivalent to P being invertible. It is wrong because it is just in fact stating (literally) that P defines an injective mapping. The second states that (11) is equivalent to (12), without the beginning of a proof. I do need to see a proof, and in particular how you "define" an invertible transform "p^-1".

* The paper then gives two examples (Sections 3, 4). In Section 3, I am a bit confused because it seems that p and q must have supports in IR, which limits the scope of the example. The same limitation applies to Section 4, even when it is a bit more interesting. In all cases, the authors must properly state a Lemma in each Section that states and shows what is claimed before Section 3.

* The paper then makes several experiments. Unless I am mistaken, it seems that Section 5.1 relies on a trick that does not change the support from x to y. Therefore, what is the interest of the approach in this case ? In Section 5.2, isn’t the trick equivalent to considering ICA with a larger \gamma ?

* A concern is that the paper says little about the reason why we should pick one p(y|x) instead of another one. The focus is on the identity of indiscernibles. The paper also forgets some potential drawbacks of the technique, including the fact that variance increases — the increase can be important with bad choices, which is certainly not a good thing.

---

> ### Author Response · Authors · 2018-11-22
> **response**
>
> * We don't understand the phrase "keeping the identity of indiscernibles" -- what does this mean?
>
> * The Zhang et al paper makes use of the spread divergence as part of a different study, namely how to use f-divergences to train generative models.
>
> * There is a connection between our approach and some forms of distances, such as Wasserstein. This connection is clear since the spread divergence connects f-divergences with MMD and there is a well known connection between MMD and the Wasserstein distance. However, we feel that that fleshing out the details of this connection is out of scope of this submission. In our view, perhaps the most "natural" training criterion for a probablistic model is maximum likelihood, since this is statistically efficient. In that sense, wherever possible, it is natural to want to train models using maximum likelihood. However, for models with different support to the data, this is no longer possible. What we show, however, is that in fact it *is* possible to train models using a form of maximum likelihood. Important questions (to our mind) are related to the statistical efficiency of using the spread divergence -- how accurately will the true model be recovered using spread noise as a function of the number of datapoints and form of the noise.
>
>
> * We agree that this is indeed a form of data processing inequality. This isn't a central point for the submissoin so we'll relegate this to an appendix, with reference to standard texts on this.
>
> * Our paper isn't phrased in formal mathematical proof language. However, if this is deemed important, we would be happy to phrase the results as proofs. For p^-1, we define it such that it has the property in equation 12 (original submission).
>
>
> * We've modified the notation to potentially reduce confusion here. All that is required for section 3 is that the domain of p(x) is a subset of R and the domain of q(x) is a subset of R. As we mentioned in the footnote of page 1, the results trivially generalise to multivariate (D-dimensional) distributions, so that p(x) and q(x) can be distributions with domains that are subsets of R^D. We would be happy to again formalise these results in an appendix, but would prefer that they appear in informal language in the main paper.
>
> * The derivation in 5.1 holds for all \gamma, including \gamma=0. In the \gamma=0 case, the model only has limited support. We'll add some text to emphasise this point.
>
> * How to choose one p(y|x) over another is briefly discussed at the end of the Summary. For this initial paper, we noted that in principle, one should learn spread noise that would maximally discriminate between the two distributions, whilst keeping the divergence finite. We gave some suggestion in section 3.2 how this could be found for simple Factor Analysis style distributions.
>
> The spread divergence idea opens up several interesting lines to study, including connections to other areas (Wassertein distances and MMD), how to best choose the spread noise in practice, convergence rates of parameters to the optimal parameters that generated the data as a function of the number of spread noise samples, etc. The central purpose of the paper is to raise awareness of the concept of a spread divergence and the potentially rich connections it has to other areas. These connections and studies still need to be explored.

---

### Official Review · AnonReviewer1 · 2018-11-08
**Interesting idea but the paper needs work**

**Rating:** 5
**Confidence:** 4

**Review:**

This paper proposes a way to define f-divergences for densities which may have different supports. While the idea itself is interesting and can be potentially very impactful, I feel the paper itself needs quite a bit of work before being accepted to a top venue.  The writing needs quite a bit oof polish for the motivations to clearly stand out. Also, some of the notation makes things way more confusing that it should be. Is it possible to use something other than p() for the noise distribution, since the problem itself is to distinguish between p() and q(). I understand the notational overload, but it complicates the reading unnecessarily. I have the following questions if the authors could please address:

1) The inequality of Zhang et a. (2018) that this paper uses seems to be an easy corollary of the Data Processing Inequality :https://en.wikipedia.org/wiki/Data_processing_inequality Did I miss something? Can the authors specify if that is not the case?

2) In terms of relevance to ICLR, the applications of PCA, ICA and training of NNs is clearly important. There seems to be a significant overlap of Sec 5.3 with Zhang et al. Could the authors specify what the differences are in terms of training methodology vis-a-vis Zhang et al? It seems to me these are parallel submissions with this submissions focussing more on properties of Spread Divergences and its non deep learning applications, while the training of NNs and more empirical evidence is moved to Zhang et al.

3) I am having a tough time understanding the derivation of Eq 25, it seems some steps were skipped. Can the authors please update the draft with more detail in the main text or appendix ?

4) Based on the results on PCA and ICA, I am wondering if the introduction of the spread is in some ways equivalent to assuming some sort of prior. In the PCA case, as an exercise to understand better, what happens if some other noise distribution is used ?

5) I do not follow the purpose of including the discussion on Fourier transforms. In general sec 3 seems to be hastily written. Similarly, what is sec 3.2's goal ?

6) The authors mention the analog to MMD for the condition \hat{D}(p,q)=0  \implies p =q. From sec 4, for the case of mercer spread divergence, it seems like the idea is that  "the eigenmaps of the embedding should match on the transformed domain" ? What is [a,b] exactly in context of the original problem? This is my main issue with this paper. They talk about the result without motivation/discussion to put things into context of the overall flow, making it harder than it should be for the reader. I have no doubt to the novelty, but the writing could definitely be improved.

---

> ### Author Response · Authors · 2018-11-22
> **response**
>
> 1) We agree and have relegated this point to an appendix. This isn't a critical part of our paper.
>
> 2) The submissions are related but different. The Zhang paper focusses on how to upper bound any f divergence, meaning that one can then train generative models using any f divergence. An example application in the Zhang submission is to training implicit generative models using a reverse KL divergence. However, in order to make the f-divergence well defined for an implicit model, the Zhang submission applies the spread divergence. The upper bound on f-divergences (Zhang submission) and the introduction of spread divergences (this submission) are independent contributions. The Zhang submission is a general idea of how to upper bound f-divgerences and in general does not require the idea of a spread divergence. This particular submission is a different general idea of how to define a spread divergence and does not require the idea of an upper bound on f-divergences. However, in certain applications of these two general ideas, these independent contributions can both be used, namely to train implicit models. Note also that the Zhang paper focusses on how to train using different f-divergences, in particular the reverse KL divergence, whereas this submission demonstrates training using standard forward KL divergence.
>
>
> 3)  Actually equation 25 isn't "derived" -- it simply follows from the assumption that p(x)/K(x) is in L2[a,b]. From Mercer's theorem, being in L2[a,b] means that there exists an orthogonal expansion of the form stated in equation 25. We'll add some text to make this more clear.
>
> 4)  The spread divergence isn't equivalent to a prior.  Perhaps one could think about p(x) as a "prior" and defining a joint p(x,y)=p(y|x)p(x), but we don't feel this is a useful direction to take and is more likely to confuse than provide insight. In general, the spread divergence isn't related to Bayesian methods.
>
> 5)  Section 3 is critical to understanding how to define a valid spread diverence and makes a connection between MMD and probabilistic models. The result is that the spread diverence is the probabilistic analog of MMD, at least for stationary spread diverengenes. The basic question is how to define a valid spread divergence. What we show in section 3 is that, for stationary spread diverengences, a valid divergence is defined when the Fourier Transform of the noise distribution Kernel is strictly positive. This is essentially the same requirement on the kernel in MMD and shows in a sense the formal relation between MMD and stationary spread divergences. Since the Gaussian has strictly positive Fourier Transform, then it defines a valid stationary spread divergence.
>
> Section 3.2 (and 3.1) both investigate further the properties of stationary spread divergences. We'll add additional motivation to the beginning of section 3.2 to explain that, whilst in section 2.1 we showed that adding noise will make two distributions more similar, there is a danger that adding too much spread noise will make training a model difficult. Section 3.2 explores some geometrical insight into the optimal direction that spread noise should take in order for the two distributions to be maximally distinguishable.
>
>
> 6) We've included additional motivation at the beginning of section 4. The main point is that in section 3 we showed the requirements on a stationary noise distribution to define a spread divergence (the Fourier Transform of the noise must be strictly positive). A natural question is then whether there are non-stationary noise distributions that can define a valid spread divergence. Whilst this is clearly true for discrete systems (as we show in section 2.1), section 4 addresses this question for continuous systems, answering it in the affirmitive.

---

### Meta-Review · Area_Chair1 · 2018-12-14

**Confidence:** 4
**Recommendation:** Reject

**Metareview:**

This manuscript proposes spread divergences as a technique for extending f-divergences to distributions with different supports. This is achieved by convolving with a noise distribution. This is an important topic worth further study in the community, particularly as it related to training generative models.

The reviewers and AC opinions were mixed, with reviewers either being unconvinced about the novelty of the proposed work, or expressing issues about the clarity of the presentation. Further improvement of the clarity, combined with additional convincing experiments would significantly strengthen this submission.